# COVID-19 pandemics modeling with modified determinist SEIR, social distancing, and age stratification. The effect of vertical confinement and release in Brazil

Wladimir Lyra[1]*, José-Dias do Nascimento Jr.[2,3], Jaber Belkhiria[4,5], Leandro de Almeida[2], Pedro Paulo M. Chrispim[6,7], Ion de Andrade[8,9]

1 Department of Astronomy, New Mexico State University, Las Cruces, New Mexico, United States of America, 2 Departamento de Física Teórica e Experimental, Universidade Federal do Rio Grande do Norte, Natal, Rio Grande do Norte, Brazil, 3 Harvard-Smithsonian Center for Astrophysics, Cambridge, Massachusetts, United States of America, 4 One Health Institute, School of Veterinary Medicine, University of California Davis, Davis, California, United States of America, 5 Center for Animal Disease Modeling and Surveillance, Department of Medicine & Epidemiology, School of Veterinary Medicine, University of California Davis, Davis, California, United States of America, 6 Instituto Alicerce Ensino Pesquisa e Inovação em Saúde, Rio de Janeiro, Rio de Janeiro, Brazil, 7 Hospital do Coração, Laboratório de Implementação do Conhecimento em Saúde, Paraíso, São Paulo, Brazil, 8 Secretaria de Estado da Saúde Pública/SESAP/ Cefope - Escola Técnica do SUS - ETSUS, Natal, Rio Grande do Norte, Brazil, 9 Laboratório de Inovação Tecnológica em Saúde, Natal, Rio Grande do Norte, Brazil

* wlyra@nmsu.edu

**Data Availability Statement:** The software is written in python 3.7, and is made public at https://github.com/wlyra/covid19.

## Abstract

The ongoing COVID-19 epidemics poses a particular challenge to low and middle income countries, making some of them consider the strategy of "vertical confinement". In this strategy, contact is reduced only to specific groups (e.g. age groups) that are at increased risk of severe disease following SARS-CoV-2 infection. We aim to assess the feasibility of this scenario as an exit strategy for the current lockdown in terms of its ability to keep the number of cases under the health care system capacity. We developed a modified SEIR model, including confinement, asymptomatic transmission, quarantine and hospitalization. The population is subdivided into 9 age groups, resulting in a system of 72 coupled nonlinear differential equations. The rate of transmission is dynamic and derived from the observed delayed fatality rate; the parameters of the epidemics are derived with a Markov chain Monte Carlo algorithm. We used Brazil as an example of middle income country, but the results are easily generalizable to other countries considering a similar strategy. We find that starting from 60% horizontal confinement, an exit strategy on May 1st of confinement of individuals older than 60 years old and full release of the younger population results in 400 000 hospitalizations, 50 000 ICU cases, and 120 000 deaths in the 50-60 years old age group alone. Sensitivity analysis shows the 95% confidence interval brackets a order of magnitude in cases or three weeks in time. The health care system avoids collapse if the 50-60 years old are also confined, but our model assumes an idealized lockdown where the confined are perfectly insulated from contamination, so our numbers are a conservative lower bound. Our results discourage confinement by age as an exit strategy.

**Funding:** The author(s) received no specific funding for this work.

**Competing interests:** The authors have declared that no competing interests exist.

## Introduction

The severe acute respiratory syndrome coronavirus 2 (SARS-CoV-2) outbreak has been ongoing for 5 months now [1]. Since it was first reported in Dec 2019 in China [2], the virus rapidly made its way to other parts of the world taking pandemic proportions [3]. The number of cases and deaths exponentially increased reaching a total of 1.5 million confirmed cases and 88 thousand deaths in early April 2020. Recent disease outbreaks that spilled over from animals such as Ebola [4, 5] or avian influenza [6] have been described as specific to developing countries. COVID-19 has been breaking this myth as the virus has been particularly exceptional at breaching in inside developed countries and challenging their health system. In Europe, Italy has been particularly affected. With 140 thousand cases, the Italian national health system has been struggling to effectively respond to the exponentially increasing flow of patients in need of intensive care [7]. The United States recently surpassed China in total number of cases (420 thousand), becoming a particular hot bed in this phase of the pandemics [8]. By the time this article is published, there will likely not be a place on Earth where the virus did not cause any damage. West African countries such as Sierra Leone just reported their first cases [9] and catastrophic scenario similar to the 2016 Ebola outbreak is possible.

The threat of COVID-19 on countries that started to count cases prompted us to develop a model to describe the evolution of the epidemic and its effects on the health care system. Mathematical models are a powerful tool that proved important in previous epidemiological disasters such as the Ebola virus [10, 11], smallpox [12], or influenza [13], contributing to the understanding of the dynamics of disease and providing useful predictions about the potential transmission of a disease and the effectiveness of possible control measures, which can provide valuable information for public health policy makers [14]. SIR-type models, also known as Kermack-McKendrick model [15], consists of a set of differential equations and has been applied to a variety of infectious diseases. Although containing simplifying assumptions, SIR models have been of great help on stopping epidemics in the past by e.g. informing effective vaccination protocols [16].

Here we develop a SEIR type compartmental model for COVID-19 including both symptomatic and asymptomatic, quarantined, and hospitalized while taking into consideration differences by age groups. We also analysed the effect of confinement during a specific period of time. Contrary to similar epidemiological models, the proposed SEIR model is initiated by the first confirmed COVID-19 death. Numerical simulations of the deterministic models are compared with real numbers of the ongoing outbreak in different countries. Moreover, the deterministic framework in which we operate greatly simplifies model analysis and allows a more thorough comparison of the various intervention strategies.

In this work we focus on the case of Brazil, where the pandemics counts 16 000 confirmed cases and 800 fatalities (April 9th, 2020). The country has 35 682 ICU beds according to government data of Feb 2020 [17]. The first official SARS-CoV-2 case in Brazil was confirmed in São Paulo on February 26th and the first official COVID-19 death was reported on March 19th. Shortly after, a lockdown was enacted first in Rio de Janeiro on March 22nd, then on other regional urban centers. There is no reliable measurement of the percentage of the population that is currently in confinement; however, the number is estimated to be around 56% according to satellite data.

Given the socio-economic consequences of a lockdown, particularly on a middle income country, decision-makers are considering a *vertical confinement* as an exit strategy to the regular lockdown. Vertical confinement is understood as reducing contact to a specific age group that is more at risk of contracting and developing SARS-CoV-2 [18], as opposed to *horizontal* (or general) confinement that does not discriminate between age groups. In the next section we will present the model, followed by validation. We then apply the model to the specific

SARS-CoV-2 scenario in Brazil, and run a sensitivity analysis. Finally, we test the effect of both general and vertical confinement on the epidemic curve.

## The model

We used a modified version of a SEIR-type deterministic compartmental model to trace COVID-19 epidemic evolution in an isolated population of $N$ individuals. We assumed that a population could be subdivided into the following compartments:

- Susceptible ($S$): COVID-19 naive individuals,

- Confined ($C$): subset of susceptibles removed from the epidemics (by e.g. social distancing).

- Exposed ($E$): Susceptible that have been exposed to infective individuals,

- Asymptomatic ($A$): Infected and infective but showing mild or no symptoms

- Symptomatic ($I$): Infected and infective but showing symptoms described in the literature,

- Quarantined ($Q$): Symptomatic that are not infective,

- Hospitalized ($H$) Symptomatic, not infective, who are being treated,

- Removed ($R$) People removed from the epidemic dynamics by recovering or passing away.

We split the population in subcategories by age (range, 0-10, 10-20, 20-30, 30-40, 40-50, 50-60, 60-70, 70-80, and 80+ years old) and we consider that some flow rates between compartments should vary with age [18].

Taking into consideration the 8 compartments and the 9 age groups, the model is described by a set of 72 coupled non-linear equations:

$$\frac{dS_i}{dt} = -\lambda(t)S_i - \psi_i(t)S_i + \phi_i(t)C_i, \tag{1}$$

$$\frac{dC_i}{dt} = \psi_i(t)S_i - \phi_i(t)C_i, \tag{2}$$

$$\frac{dE_i}{dt} = \lambda(t)S_i - \sigma E_i, \tag{3}$$

$$\frac{dA_i}{dt} = (1-p)\sigma E_i - \theta A_i, \tag{4}$$

$$\frac{dI_i}{dt} = p\sigma E_i - \gamma I_i + (1-w)\theta A, \tag{5}$$

$$\frac{dQ_i}{dt} = \gamma I_i - \xi Q_i, \tag{6}$$

$$\frac{dH_i}{dt} = q_i \xi Q_i - \eta H_i, \tag{7}$$

$$\frac{dR_i}{dt} = w\theta A_i + (1-q_i)\xi Q_i + \eta H_i. \tag{8}$$

For each compartment $X$ the age sub-bins add up to $X \equiv \Sigma_i X_i$ and compartments are such that $S + C + E + A + I + Q + H + R = N$, with $N \equiv \Sigma_i N_i$ being the total population; $N_i$ is the population in each age bin. The software is written in python 3.7, and is made public at https://github.com/wlyra/covid19.

Eqs (1)–(8) describe a compartmentalization of the population and the flow between the compartments. Contact with infected individuals removes a fraction of the susceptible ($S$) population at a rate given by $\lambda$, referred to as infection force, making them exposed ($E$) to SARS-CoV-2. The exposed ($E$) become infectious at the rate $\sigma$; a fraction $p$ of them becoming symptomatic ($I$) and a fraction $(1 - p)$ becoming asymptomatic ($A$). The symptomatic ($I$) are removed from the infective force and become quarantined ($Q$) at a rate $\gamma$. The asymptomatic ($A$) are removed at a rate $\theta$, a fraction $w$ of them going in remission and a fraction $(1 - w)$ becoming symptomatic. A fraction $q_i$ of the quarantined ($Q$) are hospitalized at a rate $\xi$. The hospitalized ($H$) are removed at a rate $\eta$. The average fatality rate is $\mu_i$.

The timescales corresponding to $\sigma$, $\gamma$, $\theta$, $\xi$, and $\eta$ are the latent period $t_\sigma \equiv \sigma^{-1}$ the infectious interval $t_\gamma \equiv \gamma^{-1}$, the remission time $t_\theta \equiv \theta^{-1}$, the time to hospitalization $t_\xi \equiv \xi^{-1}$, and the average length of hospital stay $t_\eta \equiv \eta^{-1}$.

The infection force is driven by the infected, both symptomatic ($I$) and asymptomatic ($A$)

$$\lambda(t) \equiv \beta(t)\mathcal{I}, \tag{9}$$

where we use the shorthand notation

$$\mathcal{I} \equiv \sum_i \frac{(I_i + A_i)}{N_i}, \tag{10}$$

and $\beta$ is the infection rate, related to the reproduction number $\mathcal{R}(t)$ via

$$\mathcal{R}(t) \equiv \frac{\beta(t)}{\gamma}. \tag{11}$$

Lock-down consists of having a fraction of the susceptible population removed from the epidemic dynamic by moving them from $S_i$ to $C_i$ at a rate $\psi_i$. Similarly, lifting the lock-down is done by placing $C_i$ into $S_i$ at the rate $\phi_i$. We consider these functions to be Dirac deltas

$$\psi_i \equiv a_i \; \delta(t - t_{\text{lock}}) \tag{12}$$

$$\phi_i \equiv b_i \; \delta(t - t_{\text{lift}}) \tag{13}$$

where $t_{\text{lock}}$ and $t_{\text{lift}}$ are the time (in days) of lock-down and of lifting of the lock-down, respectively. To allow for partial demographic lock-downs, $a_i$ and $b_i$ are allowed to vary by age (e.g., 80% of the 40's age group population are confined). The flow chart between compartments is shown in Fig 1.

Other diagnostic quantities are the numbers $U_i$ of people in need of an intensive care unit (ICU) bed

$$U_i \equiv \zeta_i H_i \tag{14}$$

where $\zeta_i$ is the fraction of hospitalized patients that need critical care. Both $\zeta_i$ and the hospitalization fraction $q_i$ are age-stratified.

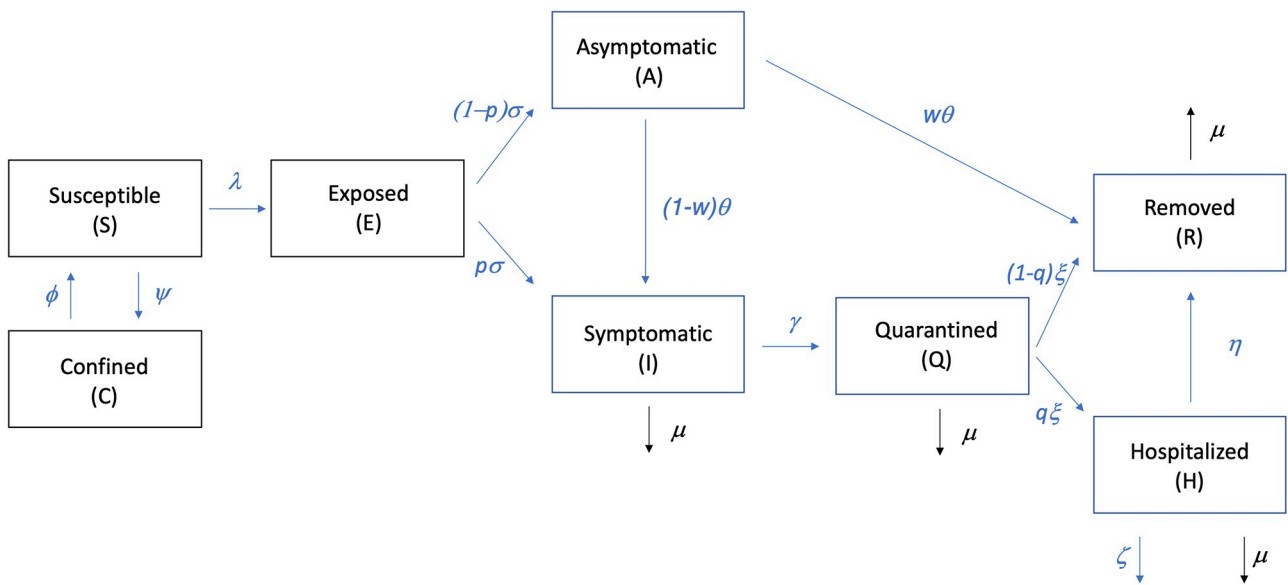

**Fig 1. Schematic flow chart between compartments.**

For integration, we use a standard Runge-Kutta algorithm, with timesteps

$$\Delta t = \frac{1}{2}[\max(\beta, \gamma, \theta, \sigma, \xi, \eta)]^{-1}. \tag{15}$$

## Model validation

In this section we present details on how we validated the model and how to determine the characteristic timescales and other parameters.

### Model fit to the 2020 COVID-19 epidemic

We consider the susceptible population ($S$) as the total population of a country since at the onset of outbreak no one is immune to the virus yet. Model parameters, shown in Table 1, were based on previous knowledge of Coronaviruses, as well as early reports and research on COVID-19 [19]. The age-dependent parameters (fatality rate $\mu_i$, fraction of infectious that are hospitalized $q_i$, and fraction of hospitalized that need critical care $\zeta_i$) are shown in Table 2.

**Table 1. Priors of timescales and ratios for the MCMC modeling.**

| Parameter | Symbol | Value | Reference |
|---|---|---:|---|
| Latent period | $\sigma^{-1}$ | 5.2 days | [19] |
| Infectious interval | $\gamma^{-1}$ | 2.9 days | [19] |
| Symptomatic fraction | $p$ | 0.6 | [18] |
| Remission time | $\theta^{-1}$ | 14 days | |
| Remission fraction of asymptomatic | $w$ | 0.8 | |
| Time to hospitalization | $\xi^{-1}$ | 5 days | [18] |
| Time at hospital | $\eta^{-1}$ | 10 days | [18] |

We use $p = 0.6$ while [18] uses $p = 2/3$. Remission time and fraction were assumed due to lack of data at the time of the study.

**Table 2. Age-dependent parameters.**

| Age bins | $\mu_i$ (×100) | $q_i$ (×100) | $\zeta_i$ (×100) |
|---:|---:|---:|---:|
| 0-10 | 0.002 | 0.1 | 5 |
| 10-20 | 0.006 | 0.3 | 5 |
| 20-30 | 0.03 | 1.2 | 5 |
| 30-40 | 0.08 | 3.2 | 5 |
| 40-50 | 0.15 | 4.9 | 6.3 |
| 50-60 | 0.60 | 10.2 | 12.2 |
| 60-70 | 2.2 | 16.6 | 27.4 |
| 70-80 | 5.1 | 24.3 | 43.2 |
| 80+ | 9.3 | 27.3 | 70.9 |

Values taken from [18].

Because all these timescales are much smaller than a human lifetime, aging of the population is ignored and no upward flow between the age sub-compartments ($i \rightarrow i + 1$) is considered. Population pyramids are taken from UN data (https://www.populationpyramid.net), and split into the pre-defined age bins.

We derive $\mathcal{R}(t)$ from the available statistics since knowledge of the real number of infected is not clear. The most reliable indicator in this situation is the number of deaths. Given a fatality rate $\mu$ and an average time $\tau$ between exposure and death, the number of dead at a time $t + \tau$ will equal the fatality rate times the number of people that got exposed at time $t$. Assuming that confinement dynamics do not play a role (although it is trivial to include it), the equation is the following:

$$\Delta D_i(t + \tau) = -\mu_i \Delta S_i(t). \tag{16}$$

Taking the continuous limit and substituting Eq (1)

$$\frac{d}{dt} D_i(t_r) = \mu_i \lambda(t) S_i \tag{17}$$

where we also write $t_r \equiv t + \tau$ for the retarded time. Summing over all age bins $D \equiv \Sigma_i D_i$ we have the cumulative death rate on the LHS, which is an observable

$$\frac{d}{dt} D(t_r) = \lambda(t) \langle \mu S \rangle \tag{18}$$

and $\langle \mu S \rangle \equiv \Sigma_i \mu_i S_i$. We can then substitute Eq (9) and solve for $\mathcal{R}(t)$ as a function of time

$$\mathcal{R}(t) = \frac{1}{\gamma \mathcal{I} \langle \mu S \rangle} \frac{d}{dt} D(t_r). \tag{19}$$

Since death occurs an average of $\tau$ days after infection, we start the integration $\tau$ days before the first reported COVID-19 death, i.e., $t = 0$ means $t_r = \tau$. The initial conditions are fully specified when the initial number of exposed individuals is defined. This should be

$$E_0(t_0) = \bar{\mu}^{-1} D_0(t_0 + \tau) \tag{20}$$

where $t_0$ is the time of the first death and $\bar{\mu} \equiv N^{-1}\sum_i \mu_i n_i$ is the age-weighted fatality rate. According to current knowledge of the epidemics, $\tau \approx 14$ days [18].

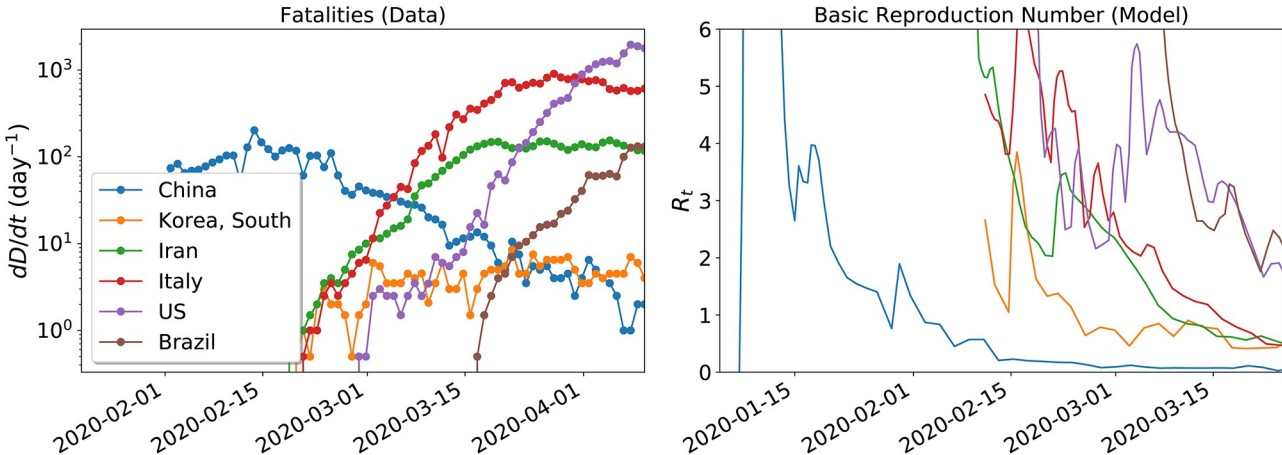

**Fig 2.** *Left*: The time series of fatalities for a number of countries. *Right*: the time derivative of the curve of fatalities is converted into $\mathcal{R}(t)$ according to Eq (19).

We compared our model predictions with official data on cases and deaths for multiple countries, as tracked by the Center for Systems Science and Engineering (CSSE) at Johns Hopkins University (https://systems.jhu.edu/research/public-health/ncov). We plot in the left panel of Fig 2 the fatality rate for a number of countries, which corresponds to the left hand side of Eq (18). We apply Eq (19) to convert this data into $\mathcal{R}(t)$, feeding this value into Eqs (1)–(8) to start the SEIR evolution. The populations $I(t)$ and $S(t)$ that enter in Eq (19) are then calculated to update $\mathcal{R}(t)$. The resulting values are plotted in the right-hand-side of Fig 2.

The timescales $\sigma$, $\gamma$, $\theta$, and $\xi$, as well as the fractions $p$ and $w$, are found by Markov chain Monte Carlo (MCMC) fitting, with the priors given in Table 1 and explained in the Supporting Information (Markov Chain Monte Carlo).

Finally, we compare the cumulative number of hospitalizations calculated from our model with the number of confirmed COVID-19 cases. For a country that is not doing massive testing and only reporting COVID-19 as acute cases reach the hospital, these curves should match reasonably well.

## Results and discussion

### Brazil epidemic scenario

Fig 3 represents the modeled epidemic scenario in Brazil up to mid-June. Parameters determined by the MCMC modeling are shown in Fig 4, being $p = 0.62^{+0.11}_{-0.13}$, $w = 0.67^{0.11}_{-0.14}$, $T_{\mathrm{inc}} = 5.42^{1.95}_{1.85}$, $T_{\mathrm{inf}} = 4.69^{2.74}_{2.32}$, $T_{\mathrm{remission}} = 13.87^{4.19}_{-5.66}$, and $T_{\mathrm{hosp}} = 8.26^{2.68}_{-3.02}$. $\mathcal{R}(t)$ at present is hovering around 2.

Fig 3a shows the evolution of the compartments of exposed ($E$), asymptomatic ($A$), symptomatic ($I$), and hospitalized ($H$), in linear scale. Fig 3b shows the same curve of $H$ but also the fraction of hospitalizations needing ICU ($U$), in log scale. The epidemic is starting at March 1st and the number of symptomatic is predicted to end at July 1st. The peak of symptomatics is predicted for May 17th with 20 million symptomatics. Consequently, there is a predicted rise in the number of hospitalized, reaching saturation on May 3rd and peaking on May 22nd with $10^6$ hospitalized. ICU beds will reach saturation on May 3rd, when the $\approx 35$ thousand ICU beds in Brazil are occupied (since the estimate assumes that all ICU beds should be occupied with coronavirus patients, which is not realistic, the collapse should in fact happen

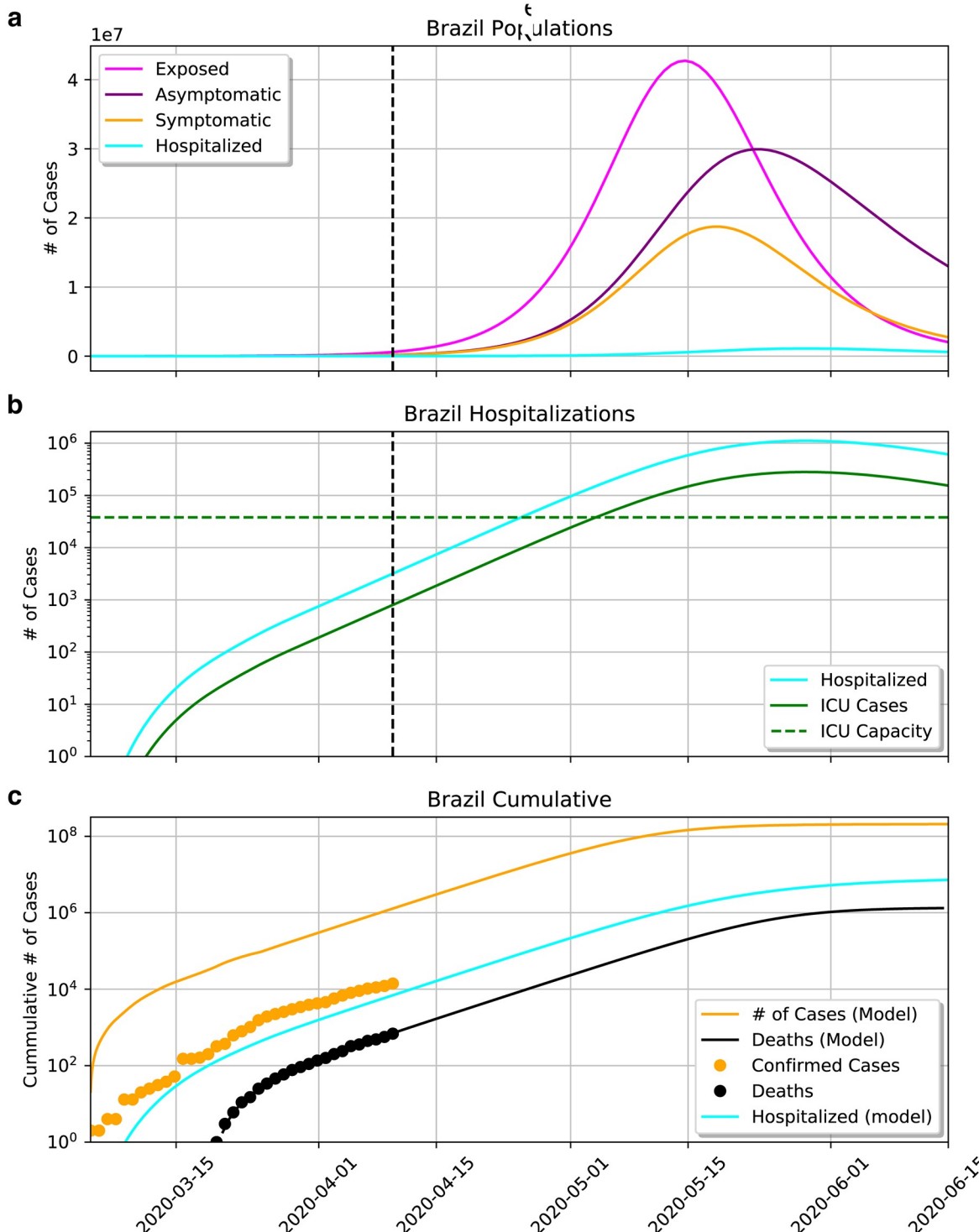

**Fig 3.** a) Evolution of the compartments of exposed (*E*), asymptomatic (*A*), symptomatic (*I*) and hospitalized (*H*). b) The same curve of *H*, and the fraction of hospitalizations needing ICU (*U*). The green dashed curve shows the total number of ICU beds in the country. At the current rate, the epidemics should peak in mid-May and collapse of the health care system should happen around May 1st. c) Cumulative number of reported cases and fatalities as orange and black dots, respectively. The number of hospitalizations closely matches the number of confirmed cases.

sooner). Demand for ICU will get higher until reaching a peak on May 22nd with 300 000 patients. The cumulative number of deaths on June 1st is $10^6$.

Fig 3c contrasts the predicted cumulative numbers of infected persons (orange line), hospitalized persons (blue line), and deaths (black line). The figure also shows the cumulative number of confirmed cases (yellow dots) and actual deaths (black dots). The cumulative number of hospitalized is very close to the actual confirmed cases. This is expected as Brazil is not doing testing on a massive scale.

We perform a sensitivity analysis, shown in Fig 5, by varying the parameters of the models by -2, 0, and 2 standard deviations as given by the results of the MCMC analysis (Fig 4). Given 7 parameters, we run $3^7 = 2187$ simulations. The fiducial model, with zero standard in all parameters, is shown as the thick line; all other models are shown as thin lines. The 95% confidence interval brackets about an order of magnitude above or below the fiducial model, or about three weeks left or right of it.

**Horizontal lockdown.** In Fig 6 we check the effect of horizontal confinement, defined as equal percentage of the population confined at any age bin. There is a change in the epidemic dynamic when horizontal confinement is applied in different rates. The plots show (a) the number of hospitalizations, (b) the number of ICU cases, and (c) the number of fatalities, as a function of the degree of social distancing. Confinement was implemented at time $t = 0$ corresponding to March 22 when the first measurement of social distancing was implemented. To not overwhelm the health care system capacity ($\approx 4 \times 10^4$) ICU beds, the level of social distancing should be over 70%. As mentioned in the introduction, estimates are that Brazil is maintaining 56% (with state-by-state variation from a maximum of 64.7% to a minimum of 53.7%). At this low level of distancing, capacity should be reached in less than 50 days, which is in agreement with the dynamical $\mathcal{R}(t)$ model in Fig 3.

**Vertical lockdown.** We vary now the degree of confinement by age bin, characterizing the vertical confinement. Fig 7 shows the number of hospitalizations in a model where confinement was implemented, broken down by age bins. The upper plots show horizontal confinement with different proportions of the population (same as Fig 6 but broken down by age and in logarithmic scale). Confinement was implemented at the same time as in Fig 6. The other rows explore vertical confinement. In the second column 60% of the population under 40 is confined, but the population older than 40 is confined to a higher degree, at 90% (solid blue line) and 99% (dashed blue line). The cyan line marks the same model as the upper plots, where 60% of the population is confined, irrespective of age. The 3rd, 4th, and 5th rows of plots show the same analysis but confining 60% of the population up 50, 60, and 70 years old, respectively. As seen in the cyan line, the number of hospitalized rises from 30 to 60 years old and falls for 70 years old onwards. That is because even though 70+ are more likely to be hospitalized, the number of 30-60 is much higher in the population.

Fig 8 shows the same results for the fraction of hospitalized that needs ICU. Fig 9 shows results from the same suite of models but for the number of fatalities. For the number of ICU cases, there is no significant difference past age 60, with only a minor uptick at the 70-80 age range. Collapse of the health care system can be avoided if vertical confinement is instored on people who are 60 or older, but at the expense of a significant number of extra ICU cases for the 50-60 age bin. At 60% confinement, hundreds of thousands of deaths are seen in the 60-70, 70-80, and 80+ age bins. The number drops to 50 000 in the 90% confinement. As noted before, vertical confinement for 60 years old and older leads to a significant number of deaths for the 50-60 age bin (over 50 000). Vertical confinement at 50 years old leads to a much lower death rate for this age segment.

Finally, we look at vertical confinement as an exit strategy. In Fig 10 we model a release from lockdown on May 1st, according to two scenarios: full release for the population under

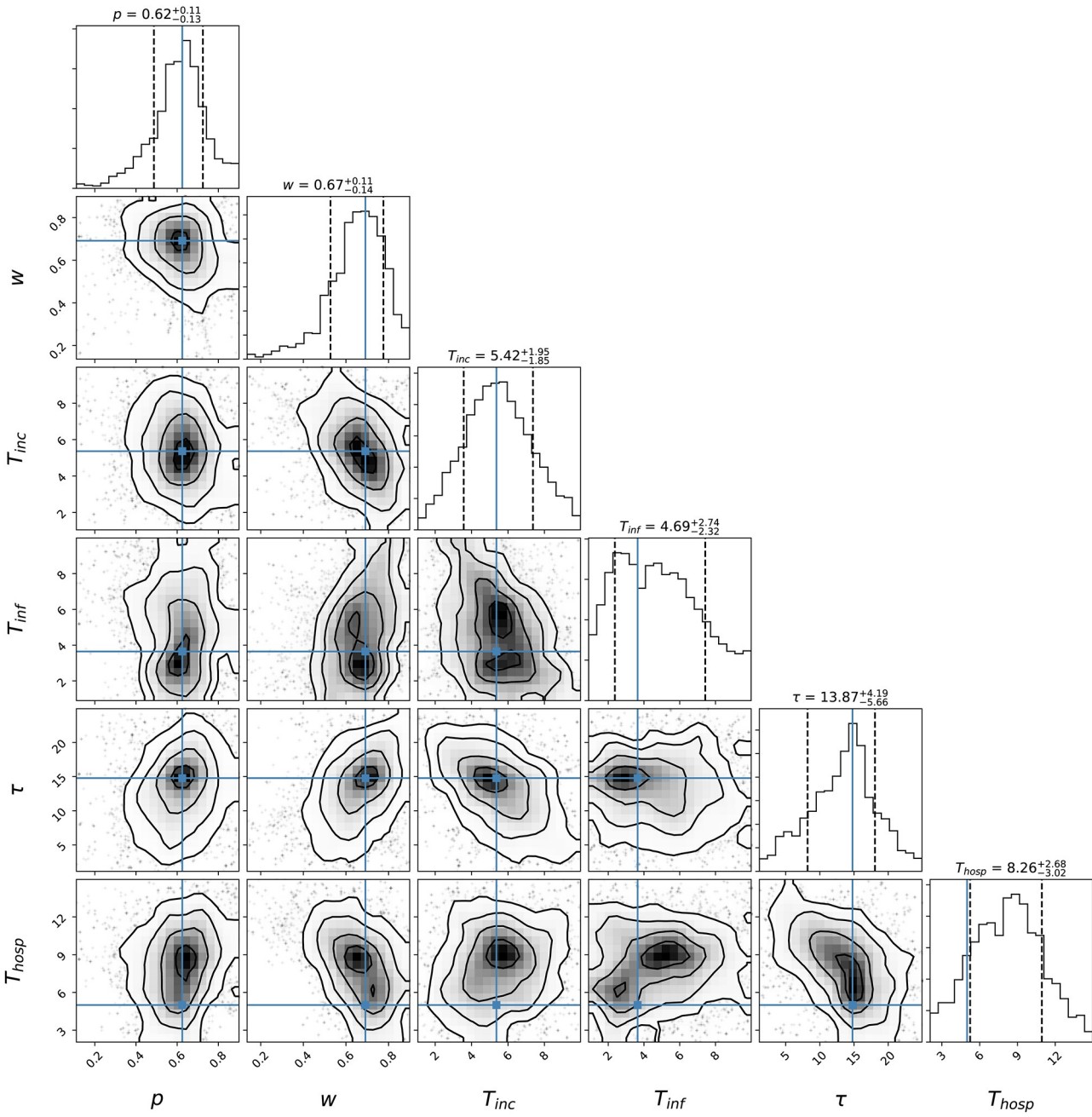

**Fig 4. Posterior probabilities for the epidemic parameters determined by the MCMC modeling.**

50 (dashed line) and full release for the population under 60 (solid line). The population past this age is kept at 90% confinement. The upper plots show the susceptible (S) and confined compartments (C), normalized by the number of individuals in the respective age bin. The second row from top to bottom shows the number of hospitalizations, the third row the number of ICU cases, and the last row the cumulative number of fatalities. As the population is released from the general confinement, the number of H/U/D peaks at 400 000/50 000/120 000 in the 50-60 age bin alone, that bears the lion's share of morbidity. Keeping the 50-60 age population in 90% confinement lowers the statistics significantly, with the health care system at capacity,

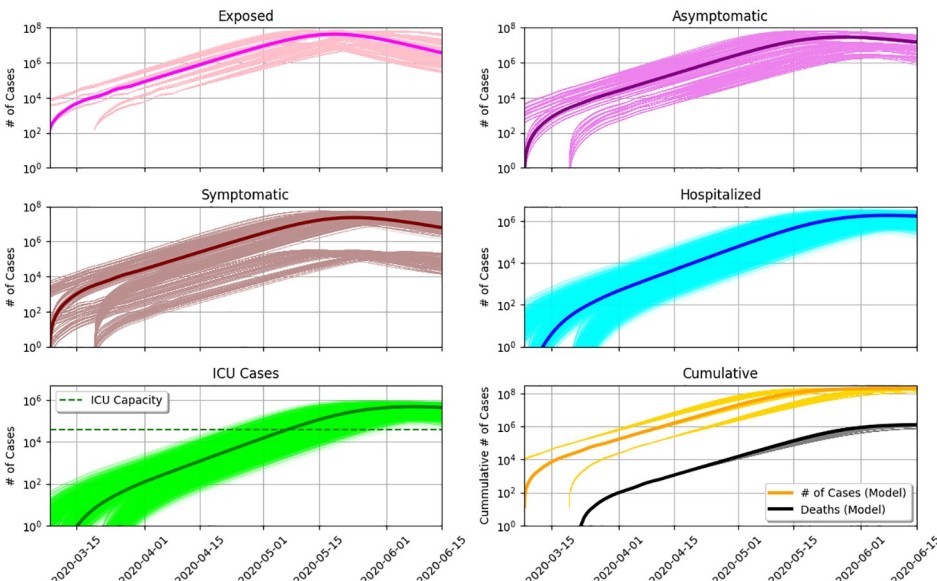

**Fig 5. The result of 2187 simulations, where the parameters used in Fig 3 are varied by -2, 0, and 2 standard deviations, as given by the MCMC analysis of Fig 4.** The model of Fig 3 (zero standard deviation on all parameters) is shown as the thick line. The 95% confidence interval brackets about an order of magnitude above or below, or about three weeks left or right of this fiducial model.

and the number of deaths per age bin about 25 000, with 60+ years olds having the same fatalities as the 40-50 age group.

## Limitations

As in any setting, the outbreak response strategy plays a crucial role on the quality of the outputs the models can give. Since the identification of the first case, the response strategy in Brazil has been changing over time. At first, only international travelers admitted to hospitals had access to SARS-CoV-2 testing. Now there are diagnostic clinics and universities involved in COVID-19 testing, but there is no national massive testing strategy in place. Besides, each Brazilian state has the authority to put in place their own strategy to address the epidemic. The states of São Paulo and Rio de Janeiro, containing the largest metropoles in the country,

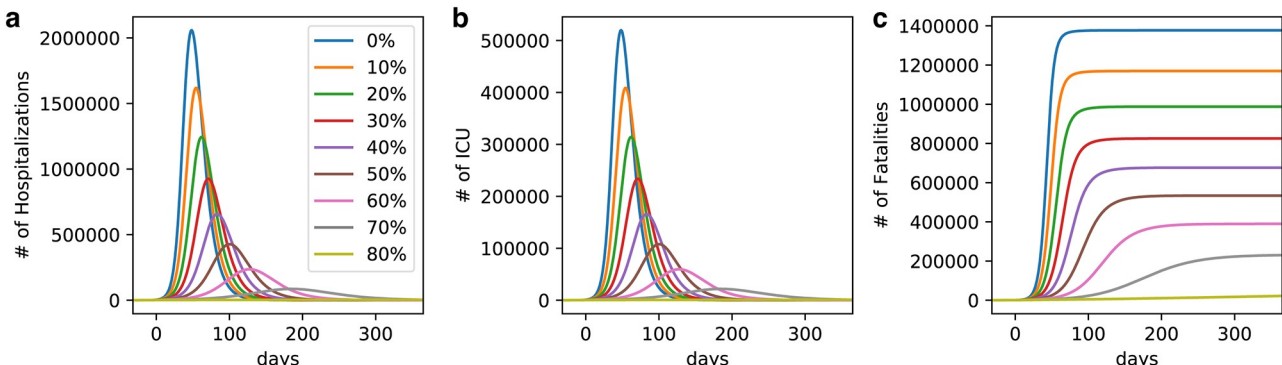

**Fig 6.** a) Number of hospitalizations. b) ICU cases. c) Fatalities. The three curves are shown as function of the degree of horizontal confinement. To not overwhelm the health care system capacity ($\approx 3 \times 10^4$) ICU beds, the level of social distancing should be over 70%. Brazil is managing 56%.

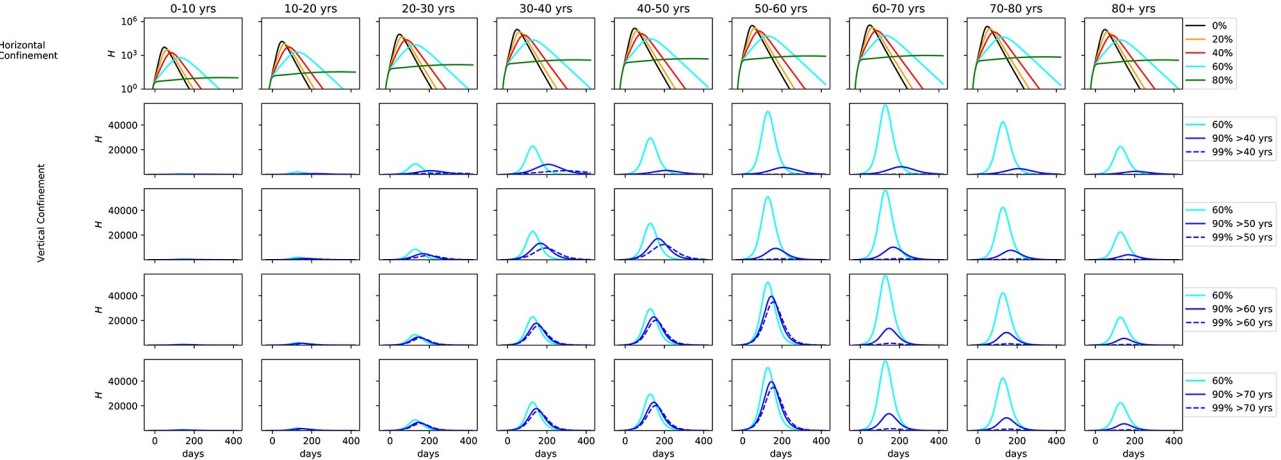

**Fig 7.** *Upper*: Number of hospitalizations in horizontal confinement with different proportions of the population, broken down by age. *2nd row*: 60% of the population under 40 is confined, the population older than 40 is confined to a higher degree, at 90% (solid blue line) and 99% (dashed blue line). The 3rd, 4th, and 5th rows of plots show the same analysis but confining 60% of the population up to 50, 60, and 70 years old, respectively.

adopted larger strategies of isolation with schools and stores closed early on while similar strategies had not yet been adopted in other states. Bottom line, the resulting morbidity and mortality rates can change significantly, resulting in dramatically different output numbers as the number of infected people or the number of hospital beds needed. It is necessary to have massive testing strategy in place to have better prediction accuracy of the models.

Our model estimate hundreds of thousands of infected people in Brazil on April 1st. This is more than the number of expected cases in the country while we write this article, considering the estimated sub-notification of cases [20] and inaction on controlling the infection. It is possible that the actual number is lower, although it is also important to notice that Brazil has not done a real lockdown so far.

The model assumes, in Eq (10) that there is no difference in infectiveness between symptomatic and asymptomatic population. This is an assumption that should be updated as further knowledge of COVID-19 is unveiled.

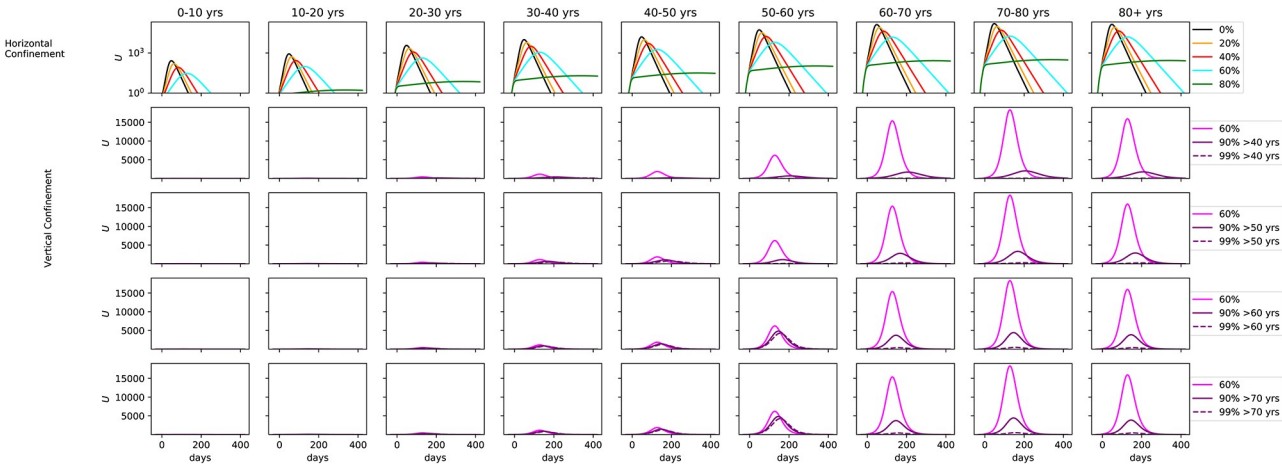

**Fig 8.** *Middle*: Same as Fig 7, but for fraction of hospitalized that need ICU.

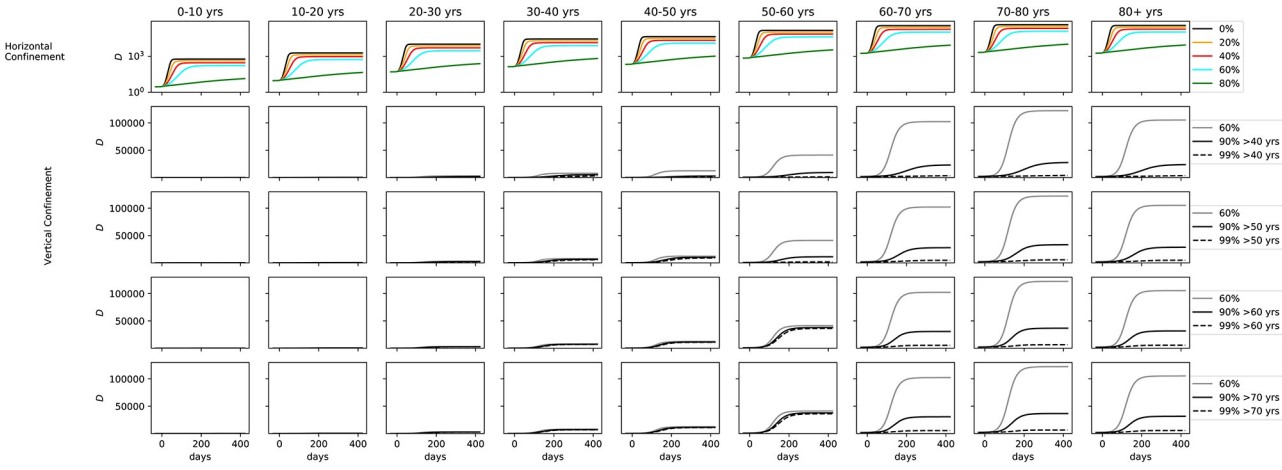

**Fig 9. Same as Fig 7, but for the number of fatalities.**

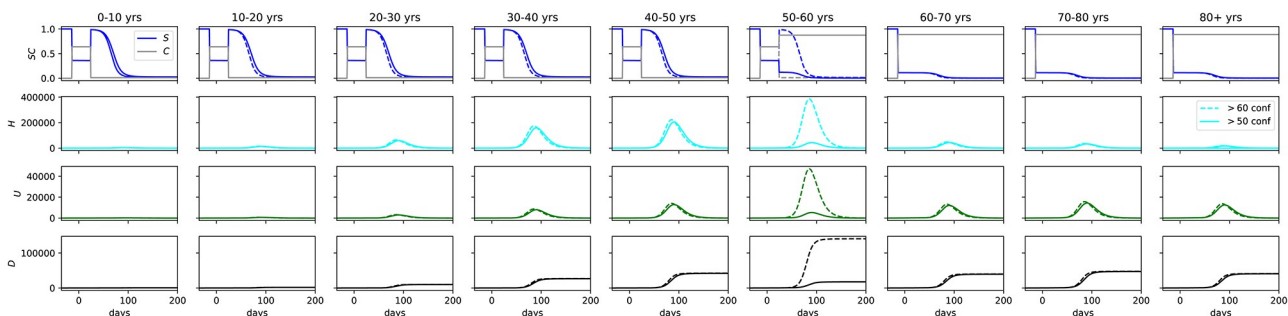

**Fig 10. Vertical confinement as exit strategy after a horizontal lockdown of 60% was held, from March 22nd to May 1st.** *Upper plots*: *S* and *C* compartments, normalized by the number of individuals in the respective age bin. *2nd row*: number of hospitalizations, *3rd row*: number of ICU cases, *lower plots*: cumulative number of fatalities. The figure shows full release for the population under 50 (dashed line) and under 60 (solid line). The population past this age is kept at 90% confinement.

We stress also that by using the cummulative hospitalization data as a guide in the MCMC, we are incurring into the problem raised by [21], of overestimating the confidence interval precision; fitting raw incidence data instead would enhance the statistical accuracy.

The model also ignores mobility, in the sense that it does not consider travel to and from the country. Given that Brazil is at the stage of community transmission (as of April 9th, 2020), this limitation should not be of significance to the results.

Conversely, and more importantly, the model assumes that the confined population is completely safe from infection, whereas in reality a vertical lockdown may not be feasible to implement as the elderly are not adequately distanced from the younger in their family and/or social circle, and infection cannot be avoided if the younger are exposed to COVID-19.

Finally, the analysis assumes that the data on fatalities is accurate. Underreported deaths should lead to an unknown source of error in the present study. Also, the MCMC produces error bars in the parameters that we did not take into account in the forward modeling.

## Conclusion

In this study we examine the strategy of vertical confinement as currently debated in Brazil. Since the fatality rate of COVID-19 appears to be higher among the elderly population, we

studied how confinement by age groups (particularly 60 years old and beyond) affects the demand for hospital beds and intensive care beds.

Our model suggests that at the current rate of advance of the pandemics, Brazil should face collapse of the health care system by May 15th, with 300 000 ICU beds needed (10 times more than the current capacity), and $10^6$ fatalities. A decrease in the rate of confirmed cases is seen with respect to the rate of fatalities, which is indicative of the effect of the lockdown. A 60% lockdown reduces the number of deaths to 400 000 due to COVID-19, still not avoiding over-load of the health care system. An increase in lockdown to 70% is needed to avoid the number of cases from overcoming the number of available critical care beds. The 95% confidence interval spans two orders of magnitude in cases or a month and a half in time.

An exit strategy of confinement of individuals older than 60 years old by May 1st would see a second wave disproportionally affect the 50-60 age bin. The ICU cases in this age range alone would bring the health care system to collapse and result in over 100 000 deaths. Confinement by age group should consider the population over 50 years old as well. However, the age range 50-60 is also a part of the workforce, and thus defeats the purpose of a confinement by age. Moreover, we emphasize that our model assumes an idealized lockdown where the confined are perfectly insulated from contamination, while in reality there would be several practical barriers to it as the confined elderly would depend on the young for most essential activities, and a perfect lockdown would not be achieved in a multi-generational household, especially in close quarters such as those found in the low and even middle income neighborhoods common in Brazil. Our results therefore discourage confinement by age as the only exit strategy. We urge Brazilian authorities to take action to prevent virus dissemination in the critical coming weeks.

# Appendix

## Markov Chain Monte Carlo

To fit the best value to $w$ and $p$, and to better constrain $\sigma^{-1}, \gamma^{-1}, \theta^{-1}, \xi^{-1}$, we use the affine-invariant ensemble sampler for Markov chain Monte Carlo (MCMC) [22] to sample the parameter space around the solutions and evaluation of the parameter uncertainties. For the priors input, we use the values taken from [18]. To search for the minimization of cumulative hospitalization $H_c$, we generated a cumulative error $C_{err}$ on the reported confirmed cases $C_c$.

As the JHU-CSSE reports on the confirmed cases are given daily with some fluctuations, we need to take this into account while weighing all solutions by adding a 1-day error matrix together with the confirmed cases (being conservative). In an ideal scenario, the cumulative number of hospitalization would be the same as the number of confirmed cases. In real life, not all confirmed cases are hospitalized so we do not expect to fit the $H_c$ with $C_c$. Rather, we weigh the $C_c$ array with the $H_c$ array using:

$$w_t = \frac{\sum C_c - H_c}{n} \tag{21}$$

$$H'_c = H_c + w_t \tag{22}$$

$H'_c$ is the weighed cumulative hospitalizations and $n$ is the length of the data. Following we get the residual between $C_c$ and $H'_c$, and we used the negative binomial distribution to calculate each likelihood [23]:

$$\text{Res} = C_c - H'_c \tag{23}$$

**Table 3. Input parameters and arrays for MCMC.**

| Parameter | Type | Value |
|---|---|---|
| $T_{inc}$: Latent period | free | 1—10 days |
| $T_{inf}$: Time while Infectious | free | 1—5 days |
| $p$: Symptomatic fraction | free | 0.2—1.0 |
| $\tau$: Days to remission after the infection | free | 6—20 days |
| $w$: Remission fraction of asymptomatic | free | 0.2—1.0 |
| $T_{hosp}$: Time to hospitalization | free | 5 days |
| Lockdown: date of the lockdown | fixed | date |
| release: date of the release of the lockdown | fixed | date |
| fatality rate age | fixed | array |
| $T_{hosp2}$: Days at hospital | fixed | 10 days |
| $q$: hospitalization fraction age | fixed | array |

Priors taken from [18].

$$Z = \frac{\sum \frac{\text{Res}}{C_{\text{err}}^2}}{\sum \frac{1}{C_{\text{err}}^2}} \tag{24}$$

$$P = \sum \frac{(\text{Res} - Z)^2}{C_{\text{err}}^2} \tag{25}$$

$$P' = -0.5P \tag{26}$$

Eqs 21 to 26 are implemented in the likelihood function on the code. If in each run it returns a finite number, the algorithm parses the result, if not it returns a large number ($10^{20}$) to discard as a bad fit.

We limit each parameter using a range cutoff in when feeding the probability function to restrict parameter space. That way, we do not run models with unrealistic physical parameters (e.g. symptomatic going to the hospital in −2 days), and also constrain the known range for the other parameters. The MCMC function feeds on 6 free parameters, 4 fixed parameters and 2 predetermined arrays as presented in Table 3.

## Acknowledgments

We thank the referee, Qianying Lin, for his constructive comments that helped improve the manuscript.

## Author Contributions

**Conceptualization:** Wladimir Lyra, Jaber Belkhiria, Pedro Paulo M. Chrispim.

**Data curation:** Wladimir Lyra.

**Formal analysis:** Wladimir Lyra, Jaber Belkhiria, Leandro de Almeida.

**Investigation:** Wladimir Lyra, José-Dias do Nascimento, Jr., Leandro de Almeida, Pedro Paulo M. Chrispim.

**Methodology:** Wladimir Lyra, José-Dias do Nascimento, Jr., Jaber Belkhiria, Leandro de Almeida, Pedro Paulo M. Chrispim, Ion de Andrade.

**Project administration:** Wladimir Lyra.

**Resources:** Wladimir Lyra.

**Software:** Wladimir Lyra, Leandro de Almeida.

**Validation:** Wladimir Lyra, Jaber Belkhiria, Leandro de Almeida, Pedro Paulo M. Chrispim, Ion de Andrade.

**Visualization:** Wladimir Lyra, Jaber Belkhiria, Leandro de Almeida, Pedro Paulo M. Chrispim.

**Writing – original draft:** Wladimir Lyra.

**Writing – review & editing:** Wladimir Lyra, José-Dias do Nascimento, Jr., Jaber Belkhiria, Leandro de Almeida, Pedro Paulo M. Chrispim, Ion de Andrade.

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
