## [Decision Letter · Decision Letter 0]

27 May 2020

PONE-D-20-10297

COVID-19 pandemics modeling with SEIR(+CAQH), social distancing, and age stratification. The effect of vertical confinement and release in Brazil.

PLOS ONE

Dear Dr. Lyra,

Thank you for submitting your manuscript to PLOS ONE. After careful consideration, we feel that it has merit but does not fully meet PLOS ONE’s publication criteria as it currently stands. Therefore, we invite you to submit a revised version of the manuscript that addresses the points raised during the review process.

We look forward to receiving your revised manuscript.

Kind regards,

Abdallah M. Samy, PhD

Academic Editor

PLOS ONE

**Journal Requirements:**

2. Please ensure that you refer to Figure 2 in your text as, if accepted, production will need this reference to link the reader to the figure.

**Reviewers' comments:**

Reviewer's Responses to Questions

**Comments to the Author**

1. Is the manuscript technically sound, and do the data support the conclusions?

Reviewer #1: Yes

2. Has the statistical analysis been performed appropriately and rigorously? 

Reviewer #1: Yes

3. Have the authors made all data underlying the findings in their manuscript fully available?

Reviewer #1: Yes

4. Is the manuscript presented in an intelligible fashion and written in standard English?

Reviewer #1: Yes

5. Review Comments to the Author

Reviewer #1: The authors present a very interesting and essential study during current COVID-19 pandemic period. City lockdown is always regarded as an effective and efficient way to cease the transmission of infectious diseases, while it would bring a huge economic loss and cause various problems including the overload of hospital beds. This study, thus, uses relatively realistic mathematical model to not only estimate the key epidemic parameters, but to evaluate the population of confinement, and to compare two strategies of confinement (i.e., horizontal and vertical). I believe this study is ready to public, though there're a few major issues which should be addressed.

1. Equation (10) is part of the force of infection, consist of both symptomatic and asymptomatic population from all age groups. However, a simple summation implies that, there's no difference in infectiveness between symptomatic and asymptomatic population. The author neither cited any reference, nor discussed this issue in limitation section.

2. Equation (11) gives the basic reproduction number. But the author failed to explain what method they used to get this result. In fact, by next generation method, I believe that some parts are missing on the right hand side.

3. Table 1 summarises the parameters used as the priors. A new column indicating the reference number would be necessary, because I found it hard to follow. For example, where are symptomatic fraction and remission fraction of asymptomatic referenced from?

4. Table 1: Incubation time is the interval between the time of being "infected" to the time of symptoms. Infectious period is the interval between the time of being "infectious" (in the case of COVID-19, it is probably before the onset of symptoms) and the time of isolation. In other words, incubation time is not sigma.

5. Line 152 suggests that the authors fitting the deterministic model to cummulative data, if I didn't get it wrong. By doing that, we could lose great variations of the data (King et al., 2015). Fitting deterministic model to incidence data may be a better choice.

6. Sensitivity analyses, or at least confidence intervals for parameters would be necessary.

6. PLOS authors have the option to publish the peer review history of their article (what does this mean?). If published, this will include your full peer review and any attached files.

Reviewer #1: Yes: Qianying Lin

---

## [Author Response · Author response to Decision Letter 0]

6 Jul 2020

Dear editor

Thank you for forwarding us the response of the referee. Our detailed answer to your requests and to the specific questions posed by the referee is shown below. In the manuscript the altered text is shown in blue colored front. 

Best wishes,

Wladimir Lyra,

Jose-Dias doNascimento Jr.

Jaber Belkhiria

Leandro de Almeida

Pedro Paulo M. Chrispim

Ion de Andrade

Editor's comments: 

Thank for you for the templates. We moved the supporting information to the end, after references, with line numbers; we also formatted the author list and addresses according to the guidelines. 

2. Please ensure that you refer to Figure 2 in your text as, if accepted, production will need this reference to link the reader to the figure.

We removed Fig 2. 

Additional edits: 

The title was changed from "COVID-19 pandemics modeling with SEIR+(CAQH), social distancing, and age stratification." to "COVID-19 pandemics modeling with modified determinist SEIR, social distancing, and age stratification."

"At the time of writing" was replaced as "as of April 9th, 2020". 

Referee's comments: 

1. Equation (10) is part of the force of infection, consist of both symptomatic and asymptomatic population from all age groups. However, a simple summation implies that, there's no difference in infectiveness between symptomatic and asymptomatic population. The author neither cited any reference, nor discussed this issue in limitation section.

The referee is correct that in the force of infection there is no difference between the symptomatic and asymptomatic, meaning that they infect at the same rate, given by the transmission coefficient beta. We now include this is the limitation section. 

2. Equation (11) gives the basic reproduction number. But the author failed to explain what method they used to get this result. In fact, by next generation method, I believe that some parts are missing on the right hand side.

It is a definition; we substitute the equal sign for the equivalent sign now.

While is true that some extra parameters go into the basic reproduction number when considering the next generation method, these extra parameters cancel out under the assumptions we make, leading to Eq 11 as we state. Given Eq 2.10 of Diekmann et al. (2010, J. R. Soc. Interface 7, 873), if one considers mu << gamma, mu << nu and p=1, we recover R_0 = beta/gamma, as we use. These approximations mean, respectively: mu << gamma: death rate mu much less than gamma, the inverse of the infectious interval. mu << gamma: death rate mu much less than nu, the inverse of the latent period. p=1 implies that there is a single latent compartment. These approximations are appropriate for our analysis.

There are other parameters that are definitions; we changed the sign from equal to equivalent throughout the text. 

3. Table 1 summarises the parameters used as the priors. A new column indicating the reference number would be necessary, because I found it hard to follow. For example, where are symptomatic fraction and remission fraction of asymptomatic referenced from?

We added the column as requested. For the latent time and infectious internal the reference

is Kucharski et al. (2020). For time to hospitalization, time at hospital, and symptomatic

fraction the reference is Fergunson et al. (2020; although we use 0.6 and they use 0.66).

For remission time and remission fraction of asymptomatic we were making assumptions.

4. Table 1: Incubation time is the interval between the time of being "infected" to the time of symptoms. Infectious period is the interval between the time of being "infectious" (in the case of COVID-19, it is probably before the onset of symptoms) and the time of isolation. In other words, incubation time is not sigma.

That is correct, sigma is the latent period. We implicitly assumed in the model that latent period and incubation time are identical. We thank the referee for pointing this out. 

5. Line 152 suggests that the authors fitting the deterministic model to cummulative data, if I didn't get it wrong. By doing that, we could lose great variations of the data (King et al., 2015). Fitting deterministic model to incidence data may be a better choice.

The referee is correct. What we are doing is to construct a diagnostic based on setting eta=0 and solving 

dH'/dt = q*xi

that is, counting how many individuals enter the hospital, and comparing that to the cummulative number

of cases. This is used to calculate posteriors. We add a sentence to the limitations stating that 

"We stress also that by using the cummulative hospitalization data

as a guide in the MCMC, we are incurring into the problem raised by King et al. (2015),

of overestimating the precision of confidence interval; fitting

raw incidence data instead would enhance the statistical accuracy."

6. Sensitivity analyses, or at least confidence intervals for parameters would be necessary.

Indeed. We wrote additional 2187 simulations, varying the parameters between -2, 0, and 2 standard deviations as

given by the MCMC analysis. The results are now shown in an extra figure. We add in the text

"We perform a sensitivity analysis, shown in Fig. 5, by varying the parameters of the models by -2, 0, and 2 standard deviations as given by the results of the MCMC analysis (Fig. 4). Given 7 parameters, we run 37 = 2187 simulations. The fiducial model, with zero standard in all parameters, is shown as the thick line; all other models are shown as thin lines. The 95% confidence interval brackets about an order of magnitude above or below the fiducial model, or about three weeks left or right of it."

and in the abstract: 

"Sensitivity analysis shows the 95\\% confidence interval brackets two orders of magnitude in cases or a month and a half in time."

---

## [Editor Report · Decision Letter 1]

31 Jul 2020

COVID-19 pandemics modeling with modified determinist SEIR, social distancing, and age stratification. The effect of vertical confinement and release in Brazil.

PONE-D-20-10297R1

Dear Dr. Lyra,

We’re pleased to inform you that your manuscript has been judged scientifically suitable for publication and will be formally accepted for publication once it meets all outstanding technical requirements.

Kind regards,

Abdallah M. Samy, PhD

Academic Editor

PLOS ONE

---

## [Editor Report · Acceptance letter]

7 Aug 2020

PONE-D-20-10297R1 

COVID-19 pandemics modeling with modified determinist SEIR, social distancing, and age stratification. The effect of vertical confinement and release in Brazil. 

Dear Dr. Lyra:

I'm pleased to inform you that your manuscript has been deemed suitable for publication in PLOS ONE. Congratulations! Your manuscript is now with our production department. 

Kind regards, 

on behalf of

Dr. Abdallah M. Samy 

Academic Editor

PLOS ONE